# Hsp60 in Skeletal Muscle Fiber Biogenesis and Homeostasis: From Physical Exercise to Skeletal Muscle Pathology

**DOI:** 10.3390/cells7120224

**Published:** 2018-11-22

**Authors:** Antonella Marino Gammazza, Filippo Macaluso, Valentina Di Felice, Francesco Cappello, Rosario Barone

**Affiliations:** 1Department of Experimental Biomedicine and Clinical Neurosciences (BioNeC), University of Palermo, 90127 Palermo, Italy; antonella.marino@hotmail.it (A.M.G.); vdfelice@inwind.it (V.D.F.); rusbarone@hotmail.it or rosario.barone@unipa.it (R.B.); 2Euro-Mediterranean Institute of Science and Technology (IEMEST), 90100 Palermo, Italy; fil.macaluso@gmail.com or filippo.macaluso1@uniecampus.it; 3Department of SMART Engineering Solutions & Technologies, eCampus University, 22060 Novedrate, Italy

**Keywords:** Hsp60, skeletal muscle, exercise, ageing, disease, regeneration, homeostasis

## Abstract

Hsp60 is a molecular chaperone classically described as a mitochondrial protein with multiple roles in health and disease, participating to the maintenance of protein homeostasis. It is well known that skeletal muscle is a complex tissue, rich in proteins, that is, subjected to continuous rearrangements, and this homeostasis is affected by many different types of stimuli and stresses. The regular exercise induces specific histological and biochemical adaptations in skeletal muscle fibers, such as hypertrophy and an increase of mitochondria activity and oxidative capacity. The current literature is lacking in information regarding Hsp60 involvement in skeletal muscle fiber biogenesis and regeneration during exercise, and in disease conditions. Here, we briefly discuss the functions of Hsp60 in skeletal muscle fibers during exercise, inflammation, and ageing. Moreover, the potential usage of Hsp60 as a marker for disease and the evaluation of novel treatment options is also discussed. However, some questions remain open, and further studies are needed to better understand Hsp60 involvement in skeletal muscle homeostasis during exercise and in pathological condition.

## 1. Hsp60 Function, Mechanisms and Interactions

The eukaryotic heat shock protein 60 (Hsp60) belongs to the group I of chaperonins, and it is classically defined as an intramitochondrial protein that assists with the correct folding of other mitochondrial proteins, together with its co-chaperonin, Hsp10 [1]. Hsp60 is essential for cell viability. Experiments were done to generate knock-outs in animal models, e.g., in mice [2,3] and in cells [4], but they were unsuccessful because the absence of Hsp60 is incompatible with life. The *Escherichia coli* Hsp60 (GroEL) and Hsp10 (GroES) are the best examined, and the information regarding structures and functions of the human counterpart come from these studies. It is well established that Hsp60 polypeptide has a mitochondrial import signal (MIS) at the N-terminus that drives Hsp60 from the cytoplasm to the mitochondria where the MIS is cleaved, and the protein reaches the final conformation [5]. When into mitochondria, Hsp60 folding and self-assembly from monomers to oligomeric species is mediated by functional pre-existing Hsp60 complexes that catalyze chaperonin folding via an ATP-dependent process [6]. The typical GroEL complex is constituted of two rings with seven identical subunits each, forming a homo-tetradecamer with a central cavity [7,8]. The GroEL subunit has three domains, apical, intermediate, and equatorial. The chaperoning function of GroEL tetradecamers requires allosteric changes and communication between subunits and rings [7]. The chaperoning mechanism is initiated by the binding of the client polypeptide to the apical domain via hydrophobic residues [9]. Then, upon the binding of ATP to the equatorial domain, the apical and intermediate domains rearrange, allowing for conformational transition (*trans* to *cis* conformation) and the encapsulation of the substrate in the central cavity. Moreover, GroES binds the same apical hydrophobic residues, and ATP hydrolysis leads to the release of the substrate into the cavity for folding. ATP hydrolysis followed by the binding of ATP to the *trans*-ring causes the dissociation of the *cis*-complex and the release of the folded protein, ADP, and GroES [7,10]. Different biophysical methods, such as light and X ray scattering, single molecule spectroscopy and hydrodynamics, have demonstrated that the human mitochondrial Hsp60 in solution seems to exist as a homo-oligomer of seven subunits, and/or in equilibrium with very minor populations of monomers and double-ring tetradecamers [11,12,13]. A single ring seems to be capable of chaperonin-mediated folding activity in vivo [14]. More recently, a study conducting using X-ray diffraction methods determined the solid state structure of the mammalian mitochondrial Hsp60–Hsp10 complex. The structure showed a symmetrical ‘football’-shaped complex of the chaperonin with the co-chaperonin [15].

From a physiological point of view, Hsp60 is constitutively expressed under normal conditions, and it can be induced in response to different kinds of stresses as a mechanism of cellular defense [16]. About one-third of cellular Hsp60 can be found in extra-mitochondrial sites, as demonstrated in many literature papers. Kirchhoff and collaborators demonstrated Hsp60 presence in the cytosol of cardiac myocytes using an antisense phosphorothioate oligonucleotide to effect a 50% reduction in the levels of mitochondrial HSP60 level [17], and atomic force and confocal microscopy experiments determined Hsp60 expression on the cell membranes of stressed human endothelial cells [18]. Moreover, Western blotting analysis showed the presence of the chaperonin in extracellular vesicles such as exosomes in a human lung mucoepidermoid cell line [19], as well as in the extracellular milieu and body fluids, using the enzyme-linked immunosorbent assay (ELISA) [20,21]. From a structural point of view, Hsp60 forms large oligomers in the other cell compartments, and functions together with Hsp10 in protein folding. Proteins that enter into the Hsp60 inner cavity are protected from interactions with other components of the surrounding environment, and as described above, Hsp10 forms a lid at the top of the double ring system, closing the opening of the central cavity [22,23]. Hsp10 is supposed to coordinate the behavior of the single Hsp60 monomers and regulate the ATPase cycle [22,24].

Recent findings have suggested that Hsp60 functions, and interactors may vary, depending on its cell and tissue localization, giving to this protein the capacity to be a multifaceted molecules in the middle of the delicate balance between health and disease. For this reason, the detection and quantitative determination of Hsp60 levels in all these locations are becoming essential components of laboratory pathology in clinics and research. In this section, we reported some examples of the intricate network around Hsp60, and of the meaning of these interactions. As stated above, inside the mitochondria, Hsp60 plays the essential role in guiding the correct folding of other mitochondrial proteins, representing an essential step in mitochondrial protein biogenesis [25,26,27]. In addition, mitochondrial Hsp60 is critically involved in the replication [28] and transmission of mitochondrial DNA (mtDNA) [29]. Moreover, it has been demonstrated that mutations in Hsp60 cause mtDNA transmission defects [29]. Recent investigations showed that Hsp60 has the capacity to unfold stable misfolded or aggregated proteins, enabling them to refold spontaneously to natively refoldable molecules [30]. The “unfoldase” activity and the role in protein folding allows Hsp60 to maintain protein homeostasis [31]. Hsp60 is also involved in intracellular protein trafficking [32] and peptide–hormone signaling [33]. Both cytosolic and mitochondrial Hsp60 isoforms play crucial roles in pro-apoptotic and pro-survival pathways [34]. Hsp60 might serve as a prognostic or diagnostic marker in certain cancers, and it can be considered as a molecular “proteus” for carcinogenesis, due to its contradictory role [16,35]. In this regard, Hsp60 upregulation in tumor cells favors cell growth, suppresses senescence, and gives resistance to stress-induced apoptosis mediating neoplastic transformation [16,36]. On the contrary, cytosolic Hsp60 may exert pro-apoptotic functions, inducing the cleavage and activation pro-caspase 3 [37]. As introduced above, Hsp60 may be present at the cell surface and in the extracellular space. Membrane-bound and extracellular Hsp60 have been shown to act as potent stimulators of immune responses [38,39]. Even in this case, there are contradictory data regarding Hsp60 role. For example, when released extracellularly, Hsp60 induces the proliferation of CD4^+^CD25^+^Foxp3 cells and suppresses cytotoxic T lymphocytes [40,41]. Moreover, increased cell surface expression of the chaperonin serves as a danger signal for the immune system, culminating in the stimulation of dendritic cells and T cell-mediated anti-tumor immune responses [42]. Extracellular Hsp60 has been demonstrated to bind to CD14, CD40, and Toll-like receptors (TLRs) such as TLR-2 and TLR-4 [43,44] A number of findings have demonstrated that extracellular Hsp60 can stimulate cellular cytokine synthesis. In this regard, Hsp60 have the capacity to induce the production of pro-inflammatory mediators like IFN-γ, TNFα, and nitric oxide, as well as IL-1, IL-6, IL-12, and IL-15 [43,45]. To the other hand, Hsp60 can stimulate the production of anti-inflammatory mediators like IL-4 and IL-10 [46,47]. Moreover, in a prospective epidemiological study of cardiovascular risk factors in healthy members of the British Civil Service, plasma Hsp60 levels have been positively associated with a decrease of cortisol, as well as with measures of psychological stress including psychological distress, job demand, and low emotional support [48]. Interestingly, the activity of Hsp60 is regulated by several post-translational modifications such as nitration [49] acetylation [21], N-glycosylation [50], and ubiquitination [51]. These modifications might impair Hsp60 structure, and/or can affect the immunological properties of the chaperonin; for example, in the tumor microenvironment [50]. All the reported data regarding the multiple activities of Hsp60 led us to draw a line between the normal and pathological condition in which the protein is contained. In our opinion, in physiological conditions, Hsp60 has the capacity to function as a homeostatic molecule, participating, for example, in the fine-tuning of inflammation, or in the maintenance of protein homeostasis, allowing the body to respond to stressful stimuli to restore tissue homeostasis. Instead, in a pathological condition, the chaperonin may function, perpetuating inflammation, and in turn favoring inflammatory chronic disease [52], autoimmune diseases [53], or cancer progression [54]. For instance, a number of papers produced in our and in other laboratories supported the idea of this dual role of the chaperonin in different systems and organs, such as the brain [20], liver [55], heart [56], lung [57], and kidney [58], just to report some examples. However, at the best of our knowledge, Hsp60 represents one of the less well-studied proteins in skeletal muscle, especially regarding physical exercise, a stressful condition that profoundly affects muscle homeostasis.

## 2. Skeletal Muscle Fiber and Adaptations

Skeletal muscle, the most abundant tissue in vertebrates, is a contractile organ responsible for body movements. It is composed by a heterogeneous population of muscle fibers, which can be classified into different types. The main classification is based on physiological, histological, or metabolic characteristics: contractile speed (slow and fast), myosin heavy chain (MHC) isoforms (from the slowest to the fastest: MHC-I, -IIa, -IIx, and -IIb), and metabolic capacity (slow oxidative, fast-oxidative and fast glycolytic) [59,60]. Three different types of skeletal muscle fibers have been identified in human (MHC-I, -IIa, and -IIx), while four are present in small rodents (MHC-I, -IIa, -IIx and -IIb) [61].

Skeletal muscle homeostasis is affected by many different types of stimuli and stresses, which can modify both gene expression and protein levels [62]. The two main stimuli that can affect skeletal muscle characteristic, inducing several types of adaptations, are physical activity [63] and nutritional modifications [64,65]. The term “physical activity” is referred to any bodily movement produced by skeletal muscles that results in energy expenditure. In particular, exercise is a subset of physical activity that is planned and structured, and if it is repetitive and has as a final aim, it can be defined as exercise training.

A single session of exercise, i.e., an acute bout of unaccustomed exercise, induces specific mechanical and metabolic stresses, which may determine muscle fiber damage. This can be observed and quantified by histologic and -omics analyses [66]. The damaged muscle fibers are repaired by adult stem cell resident in the skeletal muscle. The main source of adult stem cell in the muscle are the satellite cells, which are activated from molecules released during the damage, and can undergo differentiation to repair the damaged fibers [67]. Data in the literature suggest that an acute bout of exercise induces momentary fiber muscle damage, which stimulates an expansion of the satellite cell pool size that appears to be correlated with fiber skeletal muscle characteristics [68]. Regular exercise, i.e., exercise training, induces specific histological and biochemical adaptations, such as muscle fiber hypertrophy (increase in cross-sectional area) after strength training, or an increase of mitochondria activity and oxidative capacity after endurance training. Exercise does not induce a further increase in the expansion of the pool of satellite cells, observed after the first acute bout of exercise, it seems to remain chronically at the initial level [69,70].

A complex point to be elucidated is the role of inflammation in skeletal muscle adaptation to exercise [71,72]. While the acute inflammatory response to exercise seems to promote skeletal muscle training adaptations, persistent, low-grade inflammation, as seen in a number of chronic diseases, is obviously detrimental [73]. Although the regulation of cytokine production in skeletal muscle cells has been well investigated, little is known about the compensatory and anti-inflammatory mechanisms that resolve inflammation and that restore tissue homeostasis [71]. For example, macrophages seem to play a crucial role in impaired muscle regeneration, since these cells are associated with skeletal muscle satellite cells activation [71]. It has been proven that myocytes are able to produce a variety of immune-relevant receptors, mediators, and immunomodulatory cytokines. In this regard, damaged, mechanically stretched, as well as contracting muscle fibers, have the capacity to specifically alter the local inflammatory milieu, and thus, to attract distinct subsets of leukocytes that exert essential supportive functions in skeletal muscle remodeling, adaptation, and repair processes [74,75,76]. The inflammatory response is an inevitable consequence of myofiber damage by eccentric overload, as well as an indispensable prerequisite for subsequent structural remodeling and the functional adaptation of skeletal muscle tissue [77,78]. How this response is fine-tuned to answer to the specific demands of different exercise types is poorly understood [71].

Other conditions like ageing and several diseases cause loss of muscle strength, weakening, and muscle fiber atrophy (decrease in cross-sectional area), compromising the mobility of the subjects/patients. These conditions induce the degeneration of skeletal muscle, known as muscle wasting, which is characterized by a disarrangement of myofilament structures, and an increase in protein catabolism. Decline in muscle satellite cells number with ageing seems to be correlated with an impairment of their regenerative potential and of their self-renewal capacity (return to quiescence) [79]. These negative effects of ageing on skeletal muscle satellite cells can be reversed by exercise training in both rodents [80] and humans [81]. Figure 1 summarizes the main effects of exercise and ageing on skeletal muscle.

## 3. Hsp60 in Skeletal Muscle Fibers of Human and Small Rodent: Meaning and Implications

It is known that skeletal muscle in response to exercise produce several Hsps, and in particular, intramitochondrial Hsp60 [82,83]. In literature, the studies that investigate the levels or the expression of Hsp60 in skeletal muscle in humans are very limited. To the best of our knowledge, only two papers reported data regarding this chaperonin involvement in skeletal muscle homeostasis. Morton et al., [84] demonstrates that endurance athletes in vastus lateralis muscle have higher basal Hsp60 protein levels than sedentary subjects, although these levels did not increase after an acute bout of endurance exercise. Folkesson et al. [85] stated that Hsp60 in the vastus lateralis muscle in active and trained (resistance and endurance) subjects was not fiber type-specific. On the contrary, there is a number of papers reporting a correlation between Hsp60 and skeletal muscle adaptation using rodent models, indicating that the increased expression of Hsp60 in trained mice facilitates protein import and folding, inducing mitochondrial biogenesis [86]. Mattson et al. [87] observed that an endurance exercise training induced a significant increase in Hsp60 levels in the plantaris muscle, and no difference in the soleus muscle of trained rats. These results may be questioned, since the adaptive response to endurance exercise should be more marked in the soleus muscle, which is particularly rich in slow fibers (MHC-I) that are mainly involved during aerobic exercise. In fact, Samelman [88] observed an increase in Hsp60 levels only in the trained muscle of rats, composed mainly by type I muscle fiber (soleus) and not in muscle-rich type II muscle fibers (lateral gastrocnemius). In agreement with the previous studies, other research groups did not observe significant differences in Hsp60 levels in the plantaris and gastrocnemius muscles of rats after endurance training [89,90]. Recently, we have showed that the Hsp60 expression is fiber-specific in the gastrocnemius, soleus, and plantaris skeletal muscle of mice. Higher levels of Hsp60 were observed in type IIa, I, and IIx muscle fibers, while type IIb fibers appeared to be only slightly positive for the chaperonin [91]. Moreover, we observed an increase in Hsp60 level after endurance exercise, mainly in type I muscle fibers, and only in red gastrocnemius and soleus muscles, which are particularly rich in type I muscle fibers (Figure 2) [91]. This physiological adaptation and the increase in mitochondrial number, in response to endurance training, seems to be correlated to an increase in peroxisome proliferator-activated receptor gamma coactivator 1 alpha (PGC-1α) levels, through the phosphorylation of AMP-activated protein kinase (AMPK) [91,92]. PGC-1α is considered to be the master regulator of mitochondrial biogenesis [93].

A recent paper has reported a decrease of Hsp60 levels in a model of double knockout COX15-Alternative Oxidases myopathic skeletal muscle mouse [94]. These animals presented decreased ROS levels and impaired AMPK/PGC-1α signaling, and PAX7/MYOD-dependent muscle regeneration since the satellite cells are present, but they cannot differentiate into myotubes [94]. These findings support the idea of a positive effect of increased ROS production for mitochondrial biogenesis and muscle regeneration, and Hsp60 seems to be part of the regulation of these pathways. In agreement with these data, Ramadasan-Nair and collaborators in 2013 demonstrated that in an in vitro model of transient acute mouse muscle degeneration, Hsp60 levels decreased with the decrease of the differentiation rate of C2C12 mouse myoblast into myotubes [95]. Hsp60 has different immunological properties acting as an immune modulator and biomarker [96]. The fine-tuning of inflammation plays an essential role in skeletal muscle adaptation to exercise, as well as in tissue regeneration [71]. To the best of our knowledge, Hsp60 regulation of inflammation during/after exercise with direct effects on skeletal muscles fibers, is poorly investigated, in contrast with the well-defined role of Hsp70 [97,98]. It has been reported that physical exercise enhanced Hsp60 expression and attenuated inflammation in the adipose tissue of human diabetic obese [99]. Moreover, Hsp60 can trigger an autoimmune response in inflamed muscle tissue, being the target of regulatory autoreactive T cells, for example, in patients with juvenile dermatomyositis [100].

As stated above, other conditions such as ageing, may affect muscle strength and muscle fiber atrophy. In this regard, Wang and collaborators demonstrated that in a transgenic mouse model of premature ageing, mtDNA damage led to muscle wasting with a significant decline in muscle satellite cells [101]. The decline was accompanied by the decrease of Hsp60 and of its cognate protein, Hsp70 [101]. Hsp70 and Hsp60 play a crucial role in the replication [102] and transmission of mitochondrial DNA (mtDNA) [103], since they act as a complex, shuttling the mitochondrial transcription factor A (TFAM) to the mitochondria, as postulated by Kunkel and collaborators [104,105] All these data linking Hsp60, oxidative stress, mitochondrial biogenesis, and satellite cells indicate that Hsp60 might participate in the maintenance of skeletal muscle adaptation during exercise, and in skeletal muscle homeostasis in muscle pathology.

## 4. Hsp60 and Skeletal Muscle Pathology

The involvement of Hsp60 in skeletal muscle pathology has been investigated in various ways, from the monitoring of its quantitative levels and distribution in muscle fibers and other cells, to the immunological or genetic point of view. The diseases correlating with Hsp60 levels and structural impairment may have a direct effect on skeletal muscle homeostasis, or affect the central nervous system, and in turn, muscle activity (Table 1). Hsp60 levels decreased in the diaphragm muscle, but not in the quadriceps and gastrocnemius muscles of dystrophic-trained mice, suggesting the exhaustion of potentially protective mechanisms in the diaphragm [106]. Moreover, Hsp60 content was significantly higher in the extensor digitorum longus muscle of heat-stressed diabetic rats than in non-stressed diabetic rats, and its levels were similar to normal rat skeletal muscle [107]. Heat-stressed diabetic rats maintained Hsp60 content, resulting in the prevention of excessive ROS production and resistance to oxidative stress in the skeletal muscle [107]. An old paper from 1995, reported morphological studies indicating that Hsp60-deficient mitochondria in fibroblasts from patients are affected by mitochondrial myopathy [108]. The results were correlated to altered severely impaired mitochondrial metabolism and decreased stress tolerance [108].

From an immunological point of view, Hsp60 can trigger skeletal muscle-related diseases, inducing inflammatory and autoimmune responses. As reported above, Elst and collaborators demonstrated that human (self) Hsp60 is a disease-related autoantigen in juvenile dermatomyositis [100]. In this study, peripheral blood mononuclear cells (PBMCs) and cells obtained from muscle biopsy tissues from patients and healthy control subjects were tested for T cell proliferation induced by human and microbial Hsp60 [100]. The production of proinflammatory cytokines by muscle-derived cells in response to Hsp60 was associated with a poor clinical prognosis, whereas human Hsp60-specific induction of IL-10, TNFα, and IL-1β by PBMCs was followed by clinical remission [100]. A bioinformatics study conducted in our laboratory, showed that the chaperonin may trigger myasthenia gravis, a diseases of the neuromuscular junction, due the sequence similarity between Hsp60 from two common pathogens, *Chlamydia trachomatis* and *Chlamydia pneumoniae*, and the acetylcholine receptor α1 subunit (AChRα1) [109]. The structural data indicate that AChRα1 antibodies, implicated in the pathogenesis of myasthenia gravis, could very well be elicited and/or maintained by self- and/or bacterial Hsp60 [109]. With a similar mechanism, Hsp60 was implicated in the pathogenesis of multiple sclerosis [53]. From a genetic point of view, SPG13, an autosomal dominant form of pure hereditary spastic paraplegia, was associated with a mutation, in the gene encoding Hsp60, which results in a V72I substitution [110]. Moreover, MitCHAP-60 disease has been reported to be caused by a point mutation of Hsp60 gene at position 3 (D3G), which severely impairs its chaperone function [111]. These data led us to suppose that Hsp60 might be used as a biomarker for diagnosis and assessing prognosis, and monitoring disease status in the pathologies related to skeletal muscle impairment. Moreover, targeting Hsp60 may open up perspectives for antigen-specific immunotherapy, as well as for gene therapies.

## 5. Conclusions and Perspectives

Hsp60 is constitutively expressed in many tissues of the body in basal conditions, upon stress and under pathological conditions. A generally well-known function of Hsp60 is its involvement in the folding of mitochondrial proteins, or in the re-folding of these proteins when they are partially denatured by stress. The role of the chaperonin as a mitochondrial chaperone is relatively well known, but the properties regarding the maintenance of skeletal muscle homeostasis, both in physiological and in pathological conditions, are lacking in information. As described before, we have shown that Hsp60 increases preferentially in certain muscle fiber types, and that there is a correlation between Hsp60 up-regulation and PGC-1α expression, the master activator of the mitochondrial biogenesis. Moreover, Hsp60 has the capacity to interact with the immune system, triggering or perpetuating inflammation, a condition that influences skeletal muscle adaptation to exercise, as well as tissue regeneration. Skeletal muscle regeneration is conditioned by the satellite cells pool differentiation potential, which in turn is affected by the decrease of Hsp60 mitochondrial content, especially during ageing. Therefore, there seems to be a direct implication of the protein in the maintenance of skeletal muscle biogenesis and homeostasis, and a number of data results presented here indicate that Hsp60 also plays a crucial role in the pathogenesis of some skeletal muscle diseases. However, some questions remains open: (i) is Hsp60 a direct regulator of mitochondrial biogenesis? (ii) What are the pathways involved? (iii) What about the induction during physical exercise? (iv) In some skeletal muscle pathological conditions, should the chaperonin should be used as a biomarker or as a target for therapies? (Figure 3). Further studies are needed to better understand the Hsp60 involvement in skeletal muscle homeostasis during exercise and in pathological conditions.

## Figures and Tables

**Figure 1 cells-07-00224-f001:**
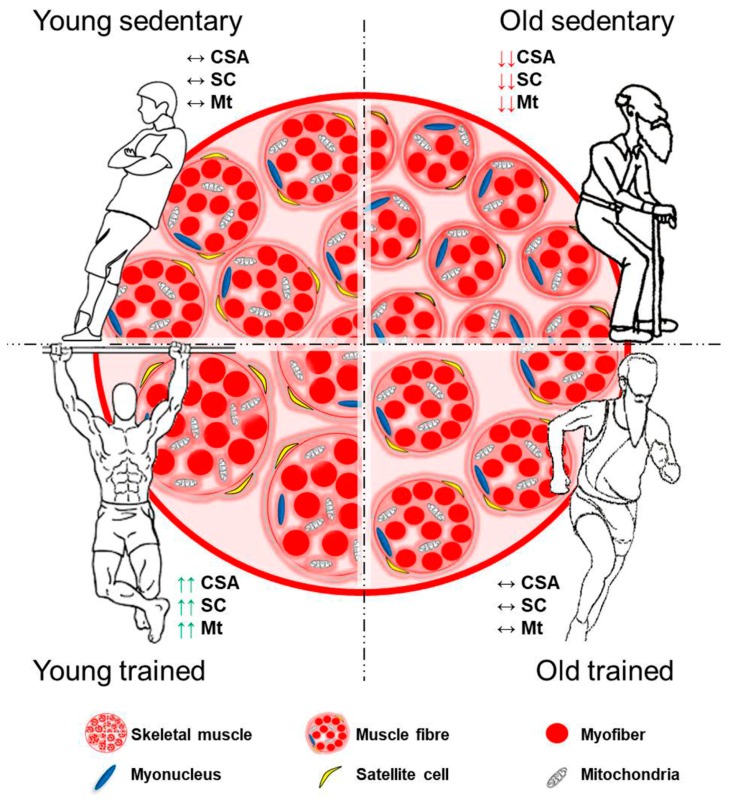
Morphological adaptation induced by physical activity and age on skeletal muscle. Skeletal muscle homeostasis is affected manly by two conditions: physical activity and age. The portrayed skeletal muscle is divided in four sections to represent four different conditions: Young sedentary, old sedentary, old trained, and young trained. Ageing cause muscle fiber atrophy (decrease in cross-sectional area, CSA) to decrease in satellite cells (SC) and mitochondrial (Mt) number. Exercise training can reverse the negative effects of ageing on skeletal muscle. Regular exercise training in young individuals induces specific adaptations, such as an increase in CSA (muscle fiber hypertrophy), SC and Mt number after strength, and/or endurance training.

**Figure 2 cells-07-00224-f002:**
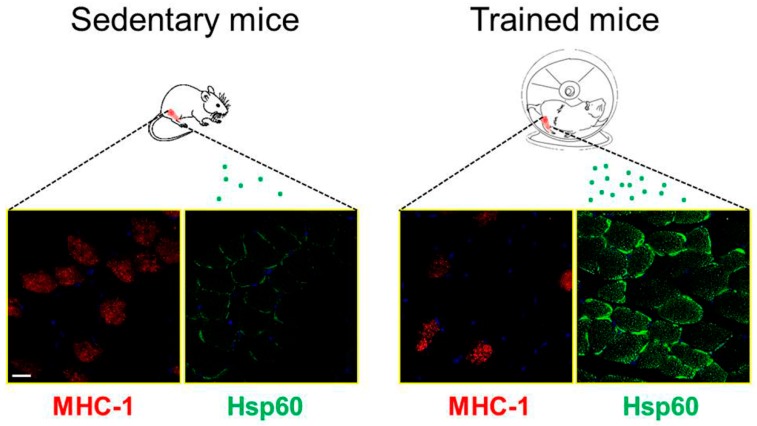
Hsp60 protein levels increase in trained mice, and they show the localization of Hsp60 in the myofibrillar cell. Immunofluorescence for Hsp60 and MHC-I (Myosin Heavy Chain–I) of cross-sections of sedentary and trained mice at 45 days in red gastrocnemius muscle. The figures show the increase of Hsp60 immunoreactivity in particular in type I muscle fibers. These fibers are particularly involved in muscle adaptation after endurance protocol training. Bar 25 μm.

**Figure 3 cells-07-00224-f003:**
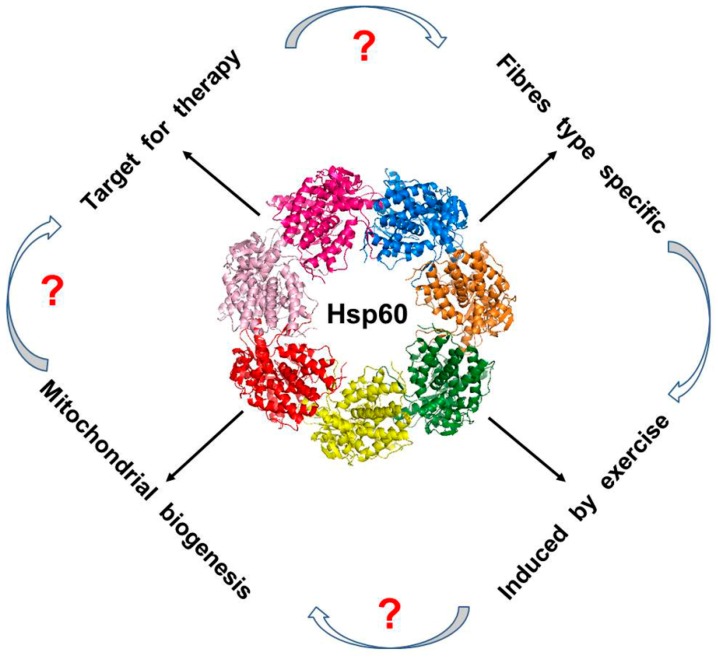
Hsp60 involvement in skeletal muscle homeostasis: working hypothesis. The direct implication of Hsp60 in the maintenance of skeletal muscle biogenesis and homeostasis has been proven by a number of papers, as reported in the text. Moreover, it seems plausible that the chaperonin might have a role in the pathogenesis of some skeletal muscle diseases. As shown in the picture, some questions remains open regarding the direct involvement in the mitochondrial biogenesis, and the meaning of its induction during exercise, as well as possible implications in some skeletal muscle diseases. In the figure a tridimensional model of Hsp60 in which each monomer is differentially colored is shown. The model was created using SWISS-MODEL [112].

**Table 1 cells-07-00224-t001:** Hsp60 and skeletal muscle pathology.

Pathologic Condition	Main Mediators	References
Dystrophic-trained mice	No mediators, Hsp60 decrease in the diaphragm muscle	[106]
Diabetic rats	No mediators, Hsp60 decrease in the extensor digitorum longus muscle	[107]
Mitochondrial myopathy	No mediators, Hsp60 deficient mitochondria in fibroblast	[108]
Dermatomyositis	T cell	[100]
Myasthenia gravis	Shared epitopes between human, *Chlamydia trachomatis* and *Chlamydia pneumoniae* Hsp60 sequence with AChRα1	[109]
Multiple sclerosis	Anti-Hsp60 *Helicobacter pylori* antibody reactivity	[53]
Hereditary spastic paraplegia	SPG13 associated with V72I substitution	[110]

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
