# Peer review of "Hsp60 in Skeletal Muscle Fiber Biogenesis and Homeostasis: From Physical Exercise to Skeletal Muscle Pathology"

_cells, 2018, doi:10.3390/cells7120224_

Round 1
Reviewer 1 Report
Thank you for the invitation to review “Hsp60 In Skeletal Muscle Fibers Biogenesis and Homeostasis: From Physical Exercise to Skeletal Muscle Pathology” by Gammazza et al. This is an interesting review article attempting to consolidate knowledge on HSP60, exploring the divergent functions in the context of skeletal muscle. Overall, this is a timely article with a novel narrative.
In the description of the extracellular role and function of HSP60, the authors cite several review articles – I feel they should refer to the original articles for these (see lines 58-70). I would recommend the inclusion of the following inclusion of J Immunol. 2005 Sep 15;175(6):3594-602 and work from Brian Henderson’s lab also Cell Stress Chaperones. 2007 Dec; 12(4): 384–392.
In description of reference 76 the authors should clarify that the study is of juvenile dermatomyositis and how they describe HSP60 as a disease-relevant auto-antigen. Furthermore, clarify the muscle-derived cells in the study.
Author Response
Here is a detailed answer to Reviewer 1.
Authors point-by-point reply to Reviewers’ Comments
Reviewer 1 comment #1 (R1C#1). In the description of the extracellular role and function of HSP60, the authors cite several review articles – I feel they should refer to the original articles for these (see lines 58-70). I would recommend the inclusion of the following inclusion of J Immunol. 2005 Sep 15;175(6):3594-602 and work from Brian Henderson’s lab also Cell Stress Chaperones. 2007 Dec; 12(4): 384–392.
Authors’ Reply (AR). In the lines 58-70 the original articles and the suggested references have been inserted, as requested by the Reviewer.
R1C#2. In description of reference 76 the authors should clarify that the study is of juvenile dermatomyositis and how they describe HSP60 as a disease-relevant auto-antigen. Furthermore, clarify the muscle-derived cells in the study.
AR. We thank the reviewer for this comment and we clarified this point in the manuscript.
Reviewer 2 Report
The chaperonin Hsp60 plays a central role for protein homeostasis in mitochondria and is therefore essential for cell viability. Although Hsp60 is expressed in all tissues, the function of this protein for muscle tissues and pathologies was so far poorly described. The review by Gammazza and colleagues provides a current overview about the role of Hsp60 for muscle biogenesis, function and pathology. The manuscript is well written and interesting. However, the review lacks a description of the well-established molecular architecture and mechanism of Hsp60/chaperonin complexes. In addition, the author should add experimental details for critical points of Hsp60 function and localization as listed below.
The authors should add to their introduction that Hsp60 is essential for cell viability. They should also briefly describe the known architecture and molecular mechanism of Hsp60 ring complexes.
The authors claim that Hsp60 promotes protein import into mitochondria (line 43). However, mitochondrial Hsp70 plays the major role in protein import, while Hsp60 performs folding of a subset of imported proteins (e.g. Chacinska et al., 2009 Cell). This point needs to be corrected.
In line 55, the authors describe that Hsp60 may catalyze cleavage of pro-caspases in apoptotic signalling. This point is misleading since Hsp60 is not a protease and need to be clarified.
Hsp60 has been found in various cell organelles (line 35). Since this is an important point for Hsp60 functions, the authors should add some information how the cellular localization was determined. Moreover, the authors should comment on the question whether Hsp60 forms large oligomers in the other cell compartments and function together with Hsp10 in protein folding?
In line 205, the authors propose that cytosolic Hsp60 promote mitochondrial targeting of TFAM. They should provide the original citation here (not a review) and describe how the role of Hsp60 in delivering of TFAM to mitochondria was determined.
The article would benefit from a table that lists all pathologies that have been linked to Hsp60.
Author Response
Here is a detailed answer to Reviewer 2.
Reviewer 2 comment #1 (R2C#1). The manuscript is well written and interesting. However, the review lacks a description of the well-established molecular architecture and mechanism of Hsp60/chaperonin complexes. In addition, the author should add experimental details for critical points of Hsp60 function and localization as listed below.
AR. We thank the reviewer for this comment and we added more information in the manuscript.
R2C#2. The authors should add to their introduction that Hsp60 is essential for cell viability. They should also briefly describe the known architecture and molecular mechanism of Hsp60 ring complexes.
AR. We added more information in the manuscript.
R2C#3. The authors claim that Hsp60 promotes protein import into mitochondria (line 43). However, mitochondrial Hsp70 plays the major role in protein import, while Hsp60 performs folding of a subset of imported proteins (e.g. Chacinska et al., 2009 Cell). This point needs to be corrected.
AR. The sentence has been re-written as suggested by the reviewer.
R2C#4. In line 55, the authors describe that Hsp60 may catalyze cleavage of pro-caspases in apoptotic signalling. This point is misleading since Hsp60 is not a protease and need to be clarified.
AR. The sentence has been re-written as suggested by the reviewer.
R2C#5. Hsp60 has been found in various cell organelles (line 35). Since this is an important point for Hsp60 functions, the authors should add some information how the cellular localization was determined. Moreover, the authors should comment on the question whether Hsp60 forms large oligomers in the other cell compartments and function together with Hsp10 in protein folding?
AR. We thank the reviewer for this comment and we added more information in the manuscript.
R2C#6. In line 205, the authors propose that cytosolic Hsp60 promote mitochondrial targeting of TFAM. They should provide the original citation here (not a review) and describe how the role of Hsp60 in delivering of TFAM to mitochondria was determined.
AR. We thank the reviewer for this comment and we added a sentence and a pertinent reference in the manuscript.
R2C#7. The article would benefit from a table that lists all pathologies that have been linked to Hsp60.
AR. We thank the reviewer for this comment. We added a table.